# Bias Estimation for Low-Cost IMU Including *X*- and *Y*-Axis Accelerometers in INS/GPS/Gyrocompass

**DOI:** 10.3390/s25051315

**Published:** 2025-02-21

**Authors:** Gen Fukuda, Nobuaki Kubo

**Affiliations:** Department of Maritime Systems Engineering, Tokyo University of Marine Science and Technology, Tokyo 135-8533, Japan; nkubo@kaiyodai.ac.jp

**Keywords:** inertial navigation systems, micro-electromechanical system, bias estimation, marine navigation

## Abstract

Inertial navigation systems (INSs) provide autonomous position estimation capabilities independent of global navigation satellite systems (GNSSs). However, the high cost of traditional sensors, such as fiber-optic gyroscopes (FOGs), limits their widespread adoption. In contrast, micro-electromechanical system (MEMS)-based inertial measurement units (IMUs) offer a low-cost alternative; however, their lower accuracy and sensor bias issues, particularly in maritime environments, remain considerable obstacles. This study proposes an improved method for bias estimation by comparing the estimated values from a trajectory generator (TG)-based acceleration and angular-velocity estimation system with actual measurements. Additionally, for *X*- and *Y*-axis accelerations, we introduce a method that leverages the correlation between altitude differences derived from an INS/GNSS/gyrocompass (IGG) and those obtained during the TG estimation process to estimate the bias. Simulation datasets from experimental voyages validate the proposed method by evaluating the mean, median, normalized cross-correlation, least squares, and fast Fourier transform (FFT). The Butterworth filter achieved the smallest angular-velocity bias estimation error. For *X*- and *Y*-axis acceleration bias, altitude-based estimation achieved differences of 1.2 × 10^−2^ m/s^2^ and 1.0 × 10^−4^ m/s^2^, respectively, by comparing the input bias using 30 min data. These methods enhance the positioning and attitude estimation accuracy of low-cost IMUs, providing a cost-effective maritime navigation solution.

## 1. Introduction

Ship position estimation generally relies on the global navigation satellite system (GNSS) in terms of cost and accuracy. However, the GNSS is vulnerable to external interference such as jamming and spoofing, and countermeasures against these threats have gained significant attention in recent years [1,2,3]. Alongside these countermeasures, the development of backup systems for situations in which the GNSS is unavailable has become a pressing need. In particular, there is a growing demand for inexpensive autonomous positioning systems resistant to external interference and capable of providing high accuracy, making them suitable for general-purpose maritime applications.

An inertial navigation system (INS), known for autonomous position estimation, does not rely on the GNSS. However, the accuracy of INSs is heavily dependent on high-cost sensors, such as fiber-optic gyroscopes (FOGs) and ring laser gyros (RLGs), which limit their deployment to specialized vessels, such as research ships and submarines. Recently, the use of low-cost inertial measurement units (IMUs) utilizing micro-electromechanical system (MEMS) technology has drawn attention. While MEMS-based IMUs offer cost advantages over FOGs or RLGs for system construction, they face challenges such as lower accuracy and significant errors caused by environmental factors [4]. Consequently, methods combining the GNSS and INS have been widely adopted to improve positioning accuracy.

Our research group aims to develop a system that can estimate positions for approximately one hour in emergency situations where the GNSS is unavailable, balancing technological, cost, and accuracy considerations. To this end, we have been investigating positioning systems that integrate MEMS-based IMUs with INS/Doppler velocity log (DVL) and INS/DVL/gyrocompass configurations [5,6]. In scenarios where the GNSS is unavailable, the sensor bias in MEMS-based IMUs significantly affects positioning accuracy, necessitating precise bias correction. This bias, representing the deviation from the true value at the start of the IMU operation, directly impacts the positioning accuracy when the GNSS is unavailable. Proper bias correction ensures accurate initial alignment of the INS and preserves system accuracy in environments where the GNSS is unavailable.

On land, techniques such as zero-velocity update (ZUPT) are commonly used to estimate IMU bias [7,8]. However, applying these techniques directly to maritime environments is challenging because of the motion of vessels, even when stationary. In visual-inertial odometry, continuous bias estimation methods utilizing deep learning have been proposed, demonstrating independence from specific movement patterns (e.g., human walking or periodic robotic motion) [9]. These characteristics suggest potential applicability to ships that exhibit non-periodic and complex movements. However, adapting such methods to maritime environments requires training models on data that reflect specific conditions, such as wave-induced accelerations and irregular vibrations, which presents a significant challenge.

Therefore, we proposed a bias estimation method using a trajectory generator (TG) applied in INS simulations [6]. This approach estimates the angular velocities and accelerations necessary to achieve the desired positioning accuracy during pure inertial navigation using data inferred by the INS/GNSS/gyrocompass (IGG). Using these estimated angular velocities and accelerations, we compared them with sensor measurements to estimate the bias. This method has the following advantages:The ability to perform stable bias estimation even under ship motion conditions.The elimination of the need to measure Earth’s rotation, enabling applicability to MEMS-based IMUs.

Previous studies show that bias estimation for the *X*- and *Y*-axis accelerations are insufficient. This issue arises when static biases exist on the *X*-, *Y*-, or both axes, leading to posture estimation errors that cannot be corrected using the Kalman filter.

When sensor values with bias in the *X*- and *Y*-axis accelerations are input into the IGG, the bias is recognized as a gravitational acceleration component owing to the tilt, causing errors in the estimated roll and pitch values that stabilize with the bias. Figure 1 compares the attitude estimates from the IGG with the reference values when the bias (0.0196 m/s^2^) is included in the *X*- and *Y*-axis acceleration components.

Although the issue of dynamic bias (random) in MEMS sensors, including accelerometers, has been widely studied, with several papers comprehensively summarizing estimation methods [10], eliminating errors in attitude estimation when static bias exists remains difficult. Methods for bias estimation using visual information have been proposed for static bias [11]. These methods are effective when landmarks are sufficiently distributed but are limited in applicability when the measurement points are in specific positional relationships or when the vessel is far from the shore. Furthermore, to conduct high-precision visual measurements of vessels, additional high-performance cameras are required, raising concerns regarding increased costs. Another approach for estimating the bias in accelerometers using attitude and angular velocity has also been considered [12]. Although this method may be applicable to vessels, it would require a system capable of estimating bias-free attitude data under the conditions we envision, necessitating additional sensors. Additionally, there are methods to estimate the bias by physically moving the IMU’s housing [13]. However, such methods require rotating platforms to move the system with high precision, which significantly increases costs. A method utilizing vessel motion during stoppage was also proposed. For example, Liu et al. [14] proposed a method for estimating the *X*- and *Y*-axis biases by comparing the accelerations in the inertial reference frame when the vessel is stopped. A similar approach, where accelerations in the inertial reference frame are compared during stoppage and a system of simultaneous equations is used to estimate the bias from multiple measurements, has also been suggested [15]. However, in previous methods, the pitch change was 6° and the roll change was 13°, which is similar to the method using simultaneous equations, requiring the vessel to oscillate while stopped. Additionally, models have been developed to estimate the acceleration bias without relying on attitude information while in motion [16]; however, these have been conducted with movements exceeding ±300° in roll and ±150° in pitch. These models provide very effective methods for estimating acceleration bias when the environment permits; however, applying them to vessels, such as the Shiojimaru training ship—which experiences minimal oscillation while moored and rarely exceeds 10° of roll or pitch while moving—is considered challenging.

During our research on bias estimation using TG, we discovered that the difference between the altitude estimates from IGG and TG correlated with the accuracy of the bias estimation. Therefore, we verified the possibility of bias estimation through a simulation by comparing the altitude estimates from IGG and TG when the static bias input to IGG was varied. Furthermore, previous methods for estimating acceleration used the closest acceleration to the target position as an indicator; however, when the trends between the measured and estimated values did not match, they remained limited to comparing averages. Therefore, we improved the method by considering the velocity estimated by the IGG, adjusting the estimated acceleration to match the trends in the measured values and examining multiple bias estimation methods, including comparisons involving averages.

This paper is organized as follows: In Section 2, the bias estimation methods for acceleration and angular velocity in this study are described in detail. Section 3 presents the simulation data generation method and the verification results of the proposed method using these data. In Section 4, the effectiveness and limitations of the proposed method are discussed. Finally, Section 5 provides the conclusions and discusses future challenges.

## 2. Estimation of Acceleration and Angular Velocity for Bias Estimation

The TG can be utilized to estimate the acceleration and angular velocity [17]; these estimates can be used for bias estimation [6]. The estimation flow, shown in Figure 2, integrates data from the IMU, GPS, and Gyrocompass using an extended Kalman filter; these integrated data are used for bias estimation.

In previous studies, velocity adjustments during the TG process were performed to approach the target position. However, in this study, the velocity is adjusted based on the trend in the IGG velocity: increasing if the IGG velocity increases or decreases if the IGG velocity decreases. The velocity closest to the target position was then adopted. This adjustment ensures that the changes in acceleration align more closely with the measured values.

Moreover, adopting estimation methods beyond simple averages is now possible. Methods such as median, cross-correlation, least squares, FFT, and Butterworth filtering have also been proposed.

Additionally, we found that the estimated biases in the *X*- and *Y*-axis accelerations correlated with the altitude differences estimated by IGG and TG. As shown in Figure 3, candidate bias values for the *X*-axis or *Y*-axis acceleration, along with the angular-velocity bias and *Z*-axis acceleration bias obtained in the TG process, were input into the IGG. The estimated values were then used to perform further estimations using the TG. At this stage, the altitude differences between the IGG and TG for each candidate bias value were recorded, and the next candidate bias value was tested. Finally, the candidate bias value that minimized the altitude difference was output as the estimated bias value.

### 2.1. Acceleration Estimation Aligned with Measured Values from Accelerometers

The velocity input into the TG was based on a geo-coordinate system. To adjust the velocity in accordance with the measured values from the accelerometers, the velocity difference in the geo-coordinate system obtained through the IGG is calculated as follows:(1)$${\underline{v}}_{{IGG}_{dif}}^{Geo}={\underline{v}}_{{IGG}_{tn}}^{Geo}-{\underline{v}}_{{IGG}_{tn-1}}^{Geo}$$
where

${\underline{v}}_{{IGG}_{tn}}^{Geo}$: velocity of IGG in the geo-coordinate system at time t_n_.

${\underline{v}}_{{IGG}_{dif}}^{Geo}$: velocity difference of IGG in the geo-coordinate system between time t_n_ and t_n−1_.

tn: the time at the sampling frequency of the TG.

In the segment, while adjusting the velocity, the dataset (angular velocity and acceleration) is determined when the IGG and TG positions are closest to each other, as shown in Figure 4.

The first line refers to the difference in acceleration in the geo-frame between times t_n_ and t_n−1_. In this study, the velocity estimation that considers the actual measurements of the accelerometer is shown from the 3rd line to the 17th line in Figure 2.

The 18th line describes the process of transforming the data from the geo-frame to the N-frame. The part indicated as TG in the figure (Line 19) performs the calculation process shown in Equations (17.2.3.1-29), (17.2.3.1-30), and (17.2.3.1-31) in Ref. [17]. $C_{B_{\left(\mathrm{tn}\right)}}^{L}$ is the attitude matrix of the TG at time tn, calculated by simultaneously using the attitudes (ϕ (roll), θ (pitch), and ψ (yaw)) in the IGG L-coordinate system [18]. Similarly, the user specifies a velocity, ${\underline{v}}_{\left(\mathrm{tn}\right)}^{N}$. As the target position is specified by the latitude and longitude, the system adjusts ${\underline{v}}_{\left(t\right)}^{\mathrm{Geo}}$ from the velocity, ${\underline{v}}_{\left(t\right)}^{\mathrm{NED}}$, in the NED coordinate system at time tn, calculated by the IGG, which is performed by adding or subtracting the amount of adjustment required to reach the desired position to ${\underline{v}}_{IGG(tn)}^{\mathrm{NED}}$ at time tn and converting it into ${\underline{v}}_{\left(tn\right)}^{\mathrm{Geo}}$ according to the definition in Ref. [18]. The calculated ${\underline{v}}_{\left(\mathrm{tn}\right)}^{N}$ is substituted into ${\underline{v}}_{m}^{N}$ in Equation (17.2.3.1-30) from Ref. [17]. The abbreviations “Angul. velo.” and “Acc.” stand for “Angular velocity” and “Acceleration”, respectively.

TG_Param represents the parameters estimated by TG and defines the dataset required for the TG calculations. TG_Param(tn − 1tn − 1) refers to the values obtained from the previous segment, including the velocity and attitude at the end of the segment. In line 20, the horizontal position difference between the IGG-estimated position and the position calculated by TG for the ii-th adjustment velocity is stored. In line 22, the system identifies the adjustment velocity where the position difference estimated by IGG and TG in the segment becomes the smallest, and the corresponding angular velocity and acceleration values are saved as values at time t_n_ in line 23.

### 2.2. Initial Bias Estimation

In this study, the estimation process was adjusted to match the increases and decreases in the acceleration estimated by the TG with the actual measurements. This process allows bias estimation by the mean value and by using methods such as the median, cross-correlation, least squares, FFT, and Butterworth.

For the acceleration along the *X*- and *Y*-axes, the bias was determined by comparing the IGG values with the altitude estimated by the TG.

#### 2.2.1. Bias Estimation Using the Mean Value

The bias estimation formula using the mean value is as follows:(2)$${\hat{B}}_{Bias}=\frac{1}{N}\sum_{i=1}^{N}\left({\underline{\tilde{\omega }}}_{{IB}_{SEN}}^{B}(i)-{\underline{\hat{\omega }}}_{{IB}_{TG}}^{B}(i)\right)$$
where

${\hat{B}}_{Bias}$: bias estimation value.

${\underline{\tilde{\omega }}}_{{IB}_{SEN}}^{B}$: angular-velocity sensor measurement.

${\underline{\hat{\omega }}}_{{IB}_{TG}}^{B}$: angular-velocity sensor estimation by TG.

N: number of data points.

#### 2.2.2. Bias Estimation Using the Median

As the bias is constant, the median of the difference between the observed data and TG-derived estimation represents the bias. Unlike the mean, the median is less sensitive to outliers or extreme values in the data. Therefore, even when measurement data contain outliers, the median is advantageous for bias estimation. In particular, in cases of significant noise or temporary anomalies, the median minimizes their influence, enabling more robust bias estimation.

The bias estimation using the median difference method can be expressed as follows:(3)$${\hat{B}}_{Bias}=median({\underline{\tilde{\omega }}}_{{IB}_{SEN}}^{B}-{\underline{\hat{\omega }}}_{{IB}_{TG}}^{B})$$
where

${\underline{\hat{\omega }}}_{{IB}_{TG}}^{B}$: estimated value.

${\underline{\tilde{\omega }}}_{{IB}_{SEN}}^{B}$: measured value by sensor.

Median: refers to the MATLAB R2024b function [19].

#### 2.2.3. Bias Estimation Using Normalized Cross-Correlation

The similarity between the bias-corrected measured signal and estimated signal was calculated using cross-correlation. The predicted bias value that yielded the highest similarity was used as the bias estimation value.

First, the measured value of the sensor ${\underline{\tilde{\omega }}}_{{IB}_{SEN}}^{B}$ is corrected by the candidate bias value $b_{n}$, as follows:(4)$${\underline{\hat{\omega }}}_{{IB}_{SEN}}^{B}(i)={\underline{\tilde{\omega }}}_{{IB}_{SEN}}^{B}(i)-b_{n}$$

Next, ${\underline{\hat{\omega }}}_{{IB}_{SEN}}^{B}(i)$ and ${\underline{\hat{\omega }}}_{{IB}_{TG}}^{B}(i)$ are normalized, as follows:(5)$${{\underline{\hat{\omega }}}_{{IB}_{SENi}}^{B}}_{Normalized}=\frac{{\underline{\hat{\omega }}}_{{IB}_{SEN}}^{B}(i)}{std({\underline{\hat{\omega }}}_{{IB}_{SEN}}^{B})},$$(6)$${{\underline{\hat{\omega }}}_{{IB}_{TGi}}^{B}}_{Normalized}=\frac{{\underline{\hat{\omega }}}_{{IB}_{TGi}}^{B}}{std({\underline{\hat{\omega }}}_{{IB}_{TGi}}^{B})},$$
where

std( ): standard deviation of ().

${{\underline{\hat{\omega }}}_{{IB}_{SEN}}^{B}}_{Normalized}$, ${{\underline{\hat{\omega }}}_{{IB}_{TG}}^{B}}_{Normalized}$: normalized sensor value and TG-estimated value.

Next, the cross-correlation between ${{\underline{\hat{\omega }}}_{{IB}_{SEN}}^{B}}_{Normalized}$ and ${{\underline{\hat{\omega }}}_{{IB}_{TG}}^{B}}_{Normalized}$ is calculated using the following equation [20,21]:(7)$$\rho (bn)=\frac{\sum_{i=1}^{N}{{\underline{\hat{\omega }}}_{{IB}_{SEN}}^{B}}_{Normalized,i}\cdot {{\underline{\hat{\omega }}}_{{IB}_{TG}}^{B}}_{Normalized,i}}{\sqrt{\sum_{i=1}^{N}{\left({{\underline{\hat{\omega }}}_{{IB}_{SEN}}^{B}}_{Normalized,i}\right)}^{2}}\cdot \sqrt{{\left({{\underline{\hat{\omega }}}_{{IB}_{TG}}^{B}}_{Normalized,i}\right)}^{2}}}$$

After calculating the correlation coefficient $\rho \left(bn\right)$ for all candidate bias values bn, the bias with the highest correlation is selected:(8)$$Num=max\left(\rho \left(bn\right)\right)$$(9)$${\hat{B}}_{Bias}=b\left(Num\right)$$

#### 2.2.4. Bias Estimation Using the Least-Squares Method

The least-squares method, which is commonly used for calculating differences, minimizes the sum of squared errors. This approach considers all data to estimate the bias.

First, the measured sensor value ${\underline{\tilde{\omega }}}_{{IB}_{SEN}}^{B}$ is corrected using a candidate bias value $b_{n}$:(10)$${\underline{\hat{\omega }}}_{{IB}_{SENi}}^{B}={\underline{\tilde{\omega }}}_{{IB}_{SENi}}^{B}-b_{n}$$

To estimate the bias bn based on the least-squares method, the following objective function J(bn) is defined:(11)$$J\left(bn\right)=\sum_{i=1}^{n}{\left[({\underline{\tilde{\omega }}}_{{IB}_{SENi}}^{B}-b_{n})-{\underline{\hat{\omega }}}_{{IB}_{TGi}}^{B}\right]}^{2}$$

The candidate bias value bn that minimizes the calculated $J\left(bn\right)$ is considered the bias estimation value.(12)$$Num=\min\left\{J\left(bn\right)\right\}$$(13)$${\hat{B}}_{Bias}=b\left(Num\right)$$

#### 2.2.5. Bias Estimation Using FFT

Bias refers to a constant offset applied across the entire signal, causing it to shift uniformly up or down in the time domain. As a result, bias appears as the DC component of the signal, which corresponds to 0 Hz in the frequency domain and can be used for bias estimation.

When the signals ${\underline{\tilde{\omega }}}_{{IB}_{SENi}}^{B}$ and $F\{{\underline{\hat{\omega }}}_{{IB}_{TGi}}^{B}\}$ are transformed into the frequency domain using FFT, their respective spectra Y(f)Y(f) and X(f)X(f) are obtained as follows:(14)$$Y(f)=F\{{\underline{\tilde{\omega }}}_{{IB}_{SEN}}^{B}\},$$(15)$$X(f)=F\{{\underline{\hat{\omega }}}_{{IB}_{TG}}^{B}\}.$$

Here, F{} represents the Fourier transform [22]. In the frequency domain, the DC components (0 Hz components) of the signals are represented by the first elements of the spectra, Y(0) and X(0), after the Fourier transform. Let the amplitudes of these components be $Y_{DC}$ and $X_{DC}$, respectively:(16)$$Y_{DC}=\frac{\left|Y(0)\right|}{N},$$(17)$$X_{DC}=\frac{\left|X(0)\right|}{N}$$
where N is the number of samples, and |Y(0)| and |X(0)| are the absolute values of the DC components of the respective spectra. The difference in the DC components of the observed signal ${\underline{\hat{\omega }}}_{{IB}_{TGi}}^{B}$ and the TG signal ${\underline{\tilde{\omega }}}_{{IB}_{SENi}}^{B}$ serves as the bias estimate. Therefore, the bias estimate ${\hat{B}}_{Bias}$ is given by(18)$${\hat{B}}_{Bias}=Y_{DC}-X_{DC}$$

#### 2.2.6. Bias Estimation Using Butterworth Filter

To estimate bias, a low-pass filter is applied to the signal to suppress high-frequency components. The transfer function of the Butterworth filter is expressed as follows [23]:(19)$$H\left(z\right)=\frac{b_{1}+b_{2}z^{-1}+\cdots +b_{r}z^{-\left(r-1\right)}}{a_{1}+a_{2}z^{-1}+\cdots +a_{r}z^{-\left(r-1\right)}}.$$

Here, b and a are vectors of filter coefficients, and n denotes the filter order. The cutoff frequency fc is set as the threshold for extracting the low-frequency components of the signal. In this study, the Butterworth filter coefficients b and a are computed using MATLAB’s butter function, as follows:(20)$$[b,a]=butter\left(order,f_{c},‘\;\mathrm{low}\;’\right).$$

Using the obtained coefficients b and a, the filter is applied to the signals ${\underline{\hat{\omega }}}_{{IB}_{TG}}^{B}(i)$ and ${\underline{\tilde{\omega }}}_{{IB}_{{SEN}_{trend}}}^{B}(i)$ to extract their trend components ${\underline{\hat{\omega }}}_{{IB}_{{TG}_{trend}}}^{B}(i)$ and ${\underline{\tilde{\omega }}}_{{IB}_{{SEN}_{trend}}}^{B}(i)$, respectively. In this program, the filtfilt [24] function in MATLAB was employed to apply a zero-phase Butterworth filter. The filter was used to extract the trend components of the signals by minimizing phase distortion and ensuring accurate bias estimation. The forward and reverse filtering operations performed by filtfilt [24] effectively removed high-frequency noise while preserving the original characteristics of the signal.(21)$${\underline{\hat{\omega }}}_{{IB}_{TG\_trendi}}^{B}=filtfilt(b,a,{\underline{\hat{\omega }}}_{{IB}_{TGi}}^{B})$$(22)$${\underline{\tilde{\omega }}}_{{IB}_{SEN\_trendi}}^{B}=filtfilt(b,a,{\underline{\tilde{\omega }}}_{{IB}_{SENi}}^{B})$$

The bias component contained in the signal ${\underline{\tilde{\omega }}}_{{IB}_{SEN\_trend}}^{B}(t)$ is estimated by calculating the difference between the trend components. The bias estimate ${\hat{B}}_{Bias}$ is computed as follows:(23)$${\hat{B}}_{Bias}=\frac{1}{N}\sum_{i=1}^{N}\left({\underline{\tilde{\omega }}}_{{IB}_{SEN\_trend}}^{B}\left(t_{i}\right)-{\underline{\hat{\omega }}}_{{IB}_{TG\_trendi}}^{B}\right)$$

### 2.3. Bias Estimation Method for X- and Y-Axis Accelerations in This Study

As shown in Figure 5, when a bias is included in the measured values of the *X*- and *Y*-axis accelerations, the estimated attitude obtained through IGG is affected by this bias, while the *Z*-axis bias is corrected. Although the estimated values may appear stable, significant altitude estimation errors occur when, for instance, the *X*-axis rotates by 90 °and aligns with the direction of gravitational acceleration. This error arises from the influence of gravitational acceleration combined with the bias. Although the altitude error can be corrected to some extent using the GNSS for *Z*-axis acceleration correction, this error still impacts the estimated values to a certain degree.

Therefore, in this study, we compared the altitude estimated by IGG with the input bias and altitude estimated by TG using these values to examine the relationship between the estimated bias and altitude estimation.

First, bias estimation for the angular velocity and *Z*-axis acceleration was performed, as described in Section 2.2.1. In the first and fourth rows of Figure 5, ${abZ}_{Bias}$ represents the angular-velocity bias and the *Z*-axis acceleration bias estimated by TG, respectively. In the third row, the predicted value for the *X*-axis acceleration bias is specified as Set Value (ii). This predicted value is determined based on bias values provided by the manufacturer and empirical values obtained from actual measurements. In the fourth row, a fixed acceleration bias value is specified for input into IGG, and the previously estimated values are assigned to ${abZ}_{Bias}$, as in the case of angular velocity. The sixth row represents the function used to calculate IGG. The seventh row illustrates the function for estimating TG using parameters obtained from IGG. In the eighth row, altitude differences from IGG and TG are compared over time, and the average of these differences is stored as the ii-th value.

In the tenth row, the ii-th bias value that minimized the average altitude difference was determined. This ii-th bias value is then defined as the *X*- or *Y*-axis acceleration bias. Although Figure 6 illustrates the processing procedure for the *X*-axis acceleration, the same processing steps were applied to estimate the bias for the *Y*-axis acceleration.

## 3. Verification of the Bias Estimation Method

The proposed bias estimation method was verified through simulations. The reference acceleration and angular velocity used in the simulations were estimated by TG based on the output values from the system installed on the Shiomijimaru. Using these simulation values, the error model of the Tamagawa Seiki IMU [25] was applied to verify the bias estimation method.

### 3.1. Simulation Values

Simulation values were derived using the attitude (roll and pitch) from the FHINS IMU manufactured by iXblue [26], heading from the gyrocompass manufactured by YDK Technologies [27], and position and velocity from the GNSS provided by TRIMBLE [28], which was configured by S-VANS. These data were processed using TG to compute accelerations and angular velocities. The error model of the IMU manufactured by Tamagawa Seiki was then applied to these simulation values to verify the bias estimation.

#### 3.1.1. Reference Data for Simulations

The reference data for simulations were obtained during experimental voyages conducted on the Shiomijimaru. Attitude data from the IMU [26], heading data from the gyrocompass [27], and position and velocity data from the Differential GNSS (DGNSS) system [28] were used to estimate accelerations and angular velocities using TG (developed by the Japan Institute of Navigation). The experimental voyage trajectory and the data segments used for the analysis are shown in the figures. Additionally, the specifications of the IMU and DGNSS are summarized in a Table 1 based on their catalog values.

Although the frequency specifications varied depending on each sensor, all data were stored at 10 Hz because of the Shiomijimaru data integration system operating at this frequency.

The reference data were estimated by inputting the above sensor data, stored in the data integration system at a 10 Hz sampling rate, into the angular velocity and acceleration estimation system [29]. The data used included the roll and pitch from the IMU, heading from the gyrocompass, and position and velocity from the GNSS.

The trajectory of the experimental voyage and the data segments used were illustrated. Additionally, Figure 6 shows the estimated values for the acceleration, angular velocity, GNSS, and horizontal errors of the simulation data.

#### 3.1.2. Specifications of Sensors Used in the Simulation

In this study, an IMU, GPS, and a gyrocompass were used. The MULTI SENSOR IMU manufactured by Tamagawa Seiki Co. Ltd., Nagano, Japan consists of MEMS-based accelerometers for the *X*-, *Y*-, and *Z*-axes and angular-velocity sensors for the *X*- and *Y*-axes. A fiber-optic gyroscope (FOG) was used as the *Z*-axis angular-velocity sensor. This configuration achieves both a low cost and high precision. Table 2 presents the catalog values provided in the specifications sheet including the sensor.

Additionally, the results of sensor measurements taken in a stationary setup in a laboratory at the Tokyo University of Marine Science and Technology (Etchujima Campus), Tokyo, Japan were analyzed using Allan variance (AV), as shown in Figure 7 and Figure 8. The Allan variance analysis was conducted for sensor modeling purposes, using IMU data sampled at 50 Hz over a duration of 44 h. The AV results and the corresponding figure are shown in Table 3 and Figure 7 and Figure 8, respectively. As a note, measurements were taken at room temperature, and the data were collected during weekends when there was minimal human traffic in the building and limited car traffic, with the data being specifically taken at night. The figures include the angle rate random walks and velocity rate random walks.

For the gyrocompass, the standard deviation was assumed to be 0.1°, as estimated using the following simplified equation:(24)$$GyroC_{simu}=round\left({Yaw}_{Ref}+rondn\cdot {GyroCom}_{STD},1\right)$$

Here, $GyroC_{simu}$ represents the simulated value of the gyrocompass; round(,1) rounds the values to the nearest hundredth to match the output of the Shiomijimaru system; and ${GyroCom}_{STD}$ is the standard deviation of the gyrocompass. The sampling frequency was set to 1 Hz, which is the same as that of the Shiomijimaru data collection system.

#### 3.1.3. Simulation Acceleration and Angular-Velocity Sensor Data

The reference values described in Section 3.1.1 were input into the error model [30] along with the values obtained from the AV, as shown in Section 3.1.2, to generate the sensor values for the simulation. For the GNSS simulation values, the reference values were input into NaveGo v1.4 [30] software, and the data were generated with a sampling frequency of 5 Hz. The bias values were set according to those listed in Table 3. Figure 9 and Figure 10 show both the measured and simulation values of angular velocity and acceleration outputs from the MULTI SENSOR, respectively. The correlations for acceleration were 0.41, 0.71, and 1.0, and the correlations for velocity were 0.95, 0.86, and 0.99, respectively. Additionally, a 1 Hz low-pass filter was applied to the acceleration on the *X*-axis, and 5 Hz low-pass filters were applied to the *Y*- and *Z*-axes. Similarly, the angular-velocity sensor data were filtered with a 5 Hz low-pass filter. The time was recorded in UTC.

### 3.2. Bias Estimation by Simulation

This section explains the bias estimation results for the angular velocity and acceleration sensors. The biases applied to the sensors were 0.2 deg/s for the *X*- and *Y*-axes of the angular-velocity sensor, 2.78 × 10^−5^ deg/s for the *Z*-axis, and 0.0196 m/s^2^ for the acceleration sensor, as specified in Table 2.

#### 3.2.1. Bias Estimation Results for Angular Velocity and Acceleration Sensors Using the TG-Based Estimation Method

For the *X*- and *Y*-axis angular-velocity sensors, the difference between the input bias values and estimated values was zero for all estimation methods. As shown Table 5, for the *Z*-axis angular velocity, the smallest difference was obtained using the Butterworth method, with a value of 2.766 × 10^−5^ deg/s. For *X*-axis acceleration, none of the methods could estimate the bias, as the estimated bias values were negative. As shown Table 6, for the *Y*-axis acceleration, the error obtained using the median method was 8.930 × 10^−3^ m/s^2^. For the *Z*-axis acceleration, the smallest error was obtained using the median method, with an error of 3.00 × 10^−5^ m/s^2^, followed by the average, least square, FFT methods, and Butterworth, which had errors of 1.700 × 10^−4^ m/s^2^. For the cross-correlation method, the correlation values were 1.0, indicating that the estimated values were identical with no difference.

#### 3.2.2. Estimation Results for *X*- and *Y*-Axis Accelerometers

The bias estimation results for the *X*- and *Y*-axis accelerometers based on the altitude differences are presented. The range of variation for the static bias during estimation was set from −0.03 to 0.03, with an increment of 0.0001. Since the values did not stabilize within 10 min, the estimation was conducted using three data segments with durations of 10, 20, and 30 min, respectively. Table 7 shows the maximum and average absolute values of the reference roll and pitch within each data segment (Seg.). Figure 11 and Figure 12 show the variations in Roll and Pitch for each segment, respectively.

In the estimation process, the static bias for *X*-axis acceleration was initially fixed at 0, while the static bias for *Y*-axis acceleration was varied to determine the value that minimized the altitude difference between IGG and TG. Subsequently, the estimated static bias for the *Y*-axis was fixed, and the static bias for *X*-axis acceleration was varied to complete the estimation. The values in parentheses in Table 8, Table 9 and Table 10 represent the fixed static bias values.

For the 20 min data with segment 1, the third iteration of estimation resulted in a difference of 3.3 × 10^−3^ m/s^2^ for the *X*-axis and 8.0 × 10^−4^ m/s^2^ for the *Y*-axis compared to the specified biases. For the 30 min data with segment 2, the third iteration yielded a difference of 1.18 × 10^−2^ m/s^2^ for the *X*-axis, while the *Y*-axis result was 1.0 × 10^−4^ m/s^2^. For the 30 min data with segment 3, the third iteration yielded a difference of 2.3 × 10^−3^ m/s^2^ for the *X*-axis, while the *Y*-axis result was 7.0 × 10^−3^ m/s^2^.

Figure 13 and Figure 14 illustrate an enlarged portion of the graph, highlighting the relationship between the altitude differences estimated by IGG and TG, and the input bias values. These figures serve as a reference for understanding how altitude differences correlate with bias estimation. These figures specifically show the results from the fourth instance of data analysis conducted on a 30 min dataset with segment 2.

The graph exhibits a parabolic shape, with the altitude difference plotted on the *Y*-axis and the input bias values on the *X*-axis. The estimated bias value was determined as the input bias value at which the altitude difference reached its minimum. This visualization highlights the correlation between the altitude differences and the estimated bias values, providing insight into the estimation process based on the specific dataset.

## 4. Discussion

No significant differences were observed among the estimation methods for the *X*- and *Y*-axes for angular-velocity bias estimation using the TG-based method. However, for the *Z*-axis angular velocity, the Butterworth filter achieved the highest accuracy, attributed to the Butterworth filter’s effectiveness as a low-pass filter.

Regarding the acceleration bias estimation using the TG-based method, no estimation method has successfully estimated the bias for the *X*-axis. For the *Y*-axis, the median filter provided the closest value to the input bias, followed by the Butterworth method. The median filter is well suited for removing spike noise and impulse noise. For the *Z*-axis, the median method yielded the most accurate value, although the differences among the methods (excluding cross-correlation) were within 10^−^⁵, rendering them negligible. Cross-correlation failed to produce meaningful results because the correlation coefficients for all estimated values were nearly 1, leaving no discernible differences. Based on these findings, the median and Butterworth filters were considered the most effective for TG-based bias estimation methods.

For the *X*- and *Y*-axis acceleration bias estimation based on the altitude difference between the IGG and TG, a correlation between the bias value and the altitude difference was confirmed. Additionally, this method provided estimates closer to the input values than the TG-based method. This study analyzed data segments of 20 and 30 min, with comparative data presented. For the *Y*-axis, the results showed that, compared to the *X*-axis, the estimates were more consistently close to the input values. Segment 3 showed a difference of 0.007 m/s^2^ compared to the input value, which was larger than those for Segments 1 and 2 and was likely influenced by the roll values. However, for the *X*-axis, the trend was not consistent, and the 20 min data provided more accurate estimates than the 30 min data, possibly attributed to coincidental results or unidentified factors that improve estimation accuracy. In any case, the estimates showed higher accuracy than those obtained using the TG-based acceleration and angular-velocity estimation method.

As shown in Figure 11 and Figure 12, the *Y*-axis was estimated more consistently because the roll values were larger than the pitch values, which resulted in better estimates. As highlighted in previous studies [12,13,14,15,16], where large roll and pitch values were used, the improvement in estimation accuracy for the *X*-axis may depend heavily on the magnitude of the pitch values.

The challenges for angular-velocity bias estimation using the TG-based method lie in reducing the estimation time and identifying navigation data that are more suitable for estimation. Nevertheless, the capability of estimating the bias during navigation is highly beneficial for position estimation in IGG, INS/DVL or INS/DVL/gyrocompass systems using low-accuracy MEMS IMUs. This capability also allows for multiple bias estimations during navigation, yielding effects similar to those of the ZUPT.

Compared to the TG-based method, the *X*- and *Y*-axis acceleration bias estimations by using altitude difference achieved higher accuracy for both axes. The roll and pitch values used in this study were smaller than those used in previous studies [12,13,14,15,16]; however, bias was successfully estimated. Furthermore, the ability to estimate bias during navigation has effects similar to those of the ZUPT.

The main challenge lies in the time required for estimation, making real-time estimation difficult under the current conditions. Considering the altitude differences are in the order of 10^−2^, a certain amount of data are necessary. Further analysis is required to determine the extent to which roll and pitch values influence the results. The correlation between altitude differences and bias values is attributed to the residual *X*- and *Y*-axis bias components in the *Z*-axis velocity of the N-frame, leading to errors in the *Z*-axis velocity. However, this method has not demonstrated, through calculation, that the altitude difference between the IGG and TG correlates with accelerometer bias, which remains a challenge for future studies.

## 5. Conclusions

This study proposes a bias estimation method utilizing a TG for accelerations and angular velocities in MEMS-based IMUs. The simulation results demonstrated the effectiveness of the Butterworth filter for angular-velocity bias estimation, which achieved the highest accuracy among all tested methods. For acceleration bias estimation, the proposed method using altitude differences between the IGG and TG proved effective for both the *X*- and *Y*-axis bias.

The findings also revealed limitations, such as the requirement for extended data collection periods and the lack of a theoretical explanation for the correlation between altitude differences and bias values. Further investigation of these aspects is essential to enhance the robustness and applicability of this method.

Future work will focus on the following aspects.
Refining the theoretical framework underlying the observed correlations.Integration of the proposed method with other navigation systems.Extending the application to real-time maritime navigation scenarios and other domains, such as aerial and terrestrial vehicles.

This achievement contributes to enhancing the accuracy of IGG and INS/GNSS systems utilizing low-cost IMUs. This research also provides a significant step toward reliable and cost-effective navigation solutions for scenarios where the GNSS is unavailable, thereby enhancing the safety and efficiency of maritime operations.

## Figures and Tables

**Figure 1 sensors-25-01315-f001:**
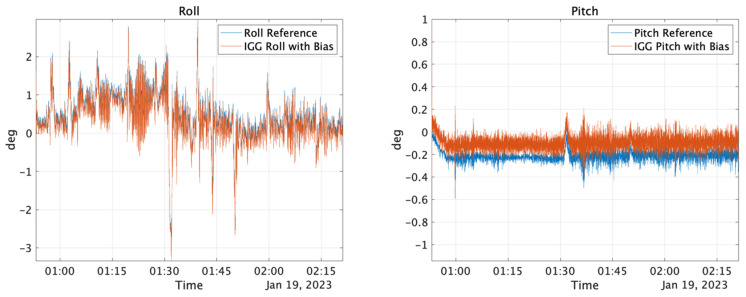
Roll and pitch comparison between reference and IGG with X and Y Acc. Initial bias.

**Figure 2 sensors-25-01315-f002:**
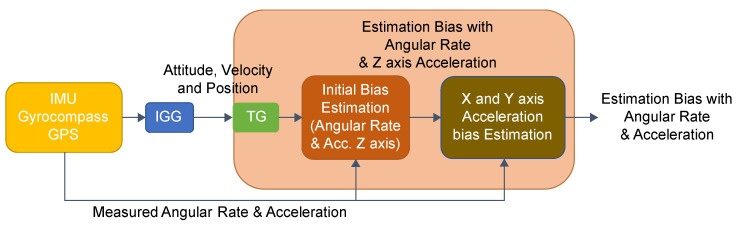
Bias estimation process.

**Figure 3 sensors-25-01315-f003:**
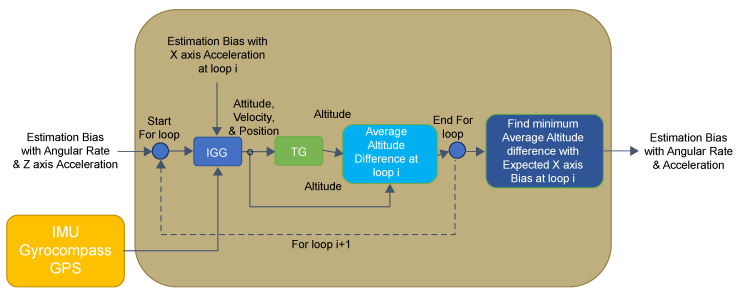
Details of the X and Y acceleration initial bias estimation section in Figure 2.

**Figure 4 sensors-25-01315-f004:**
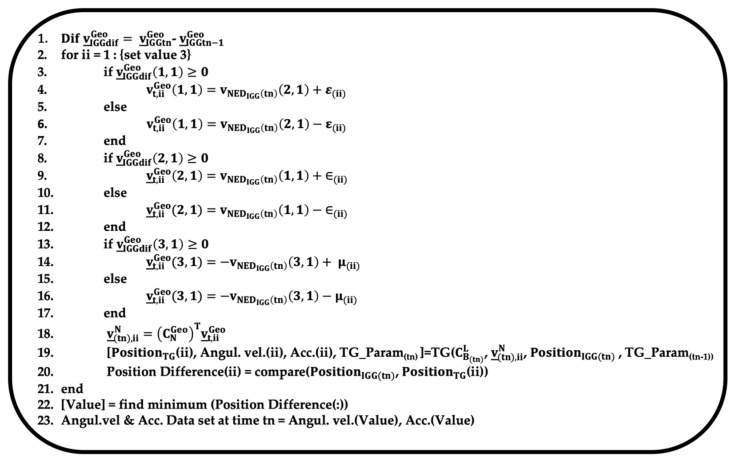
Image of processing program at a particular segment and time.

**Figure 5 sensors-25-01315-f005:**
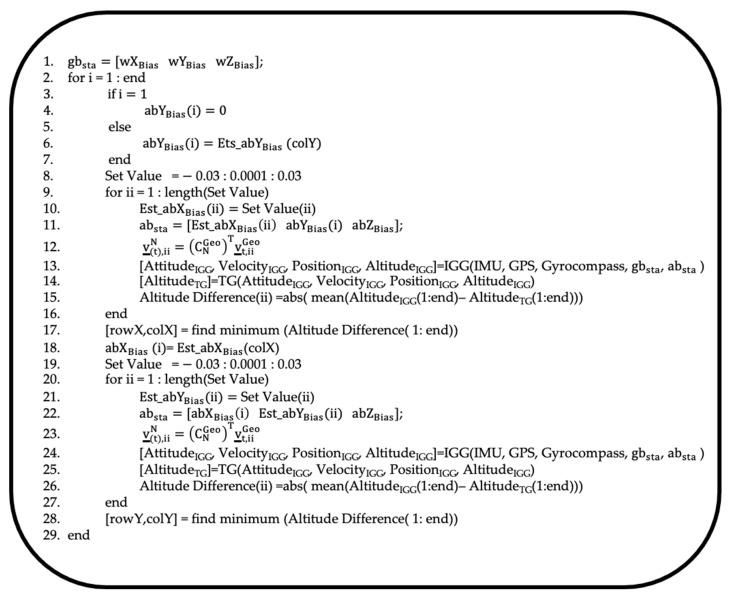
Image of processing program at a particular segment and time.

**Figure 6 sensors-25-01315-f006:**
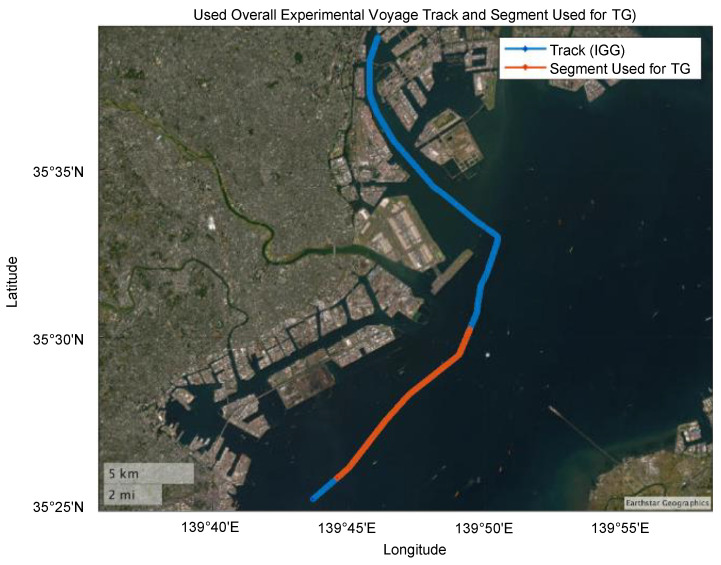
Experimental voyage track used for simulation.

**Figure 7 sensors-25-01315-f007:**
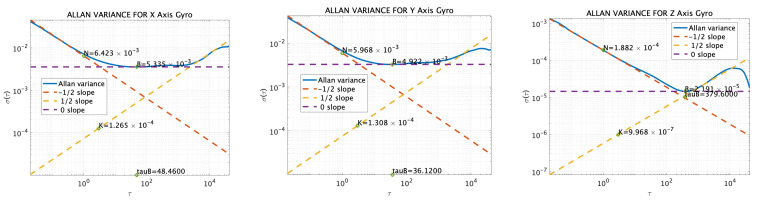
AV plots for gyroscopes.

**Figure 8 sensors-25-01315-f008:**
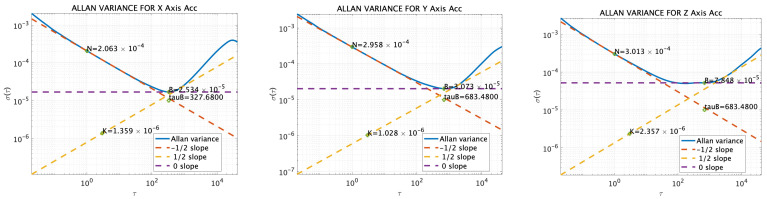
AV plots for accelerometers. For the GNSS simulation values, RTK positioning using u-blox F9P, as shown in Table 4, and NovAtel GNSS-802L is assumed.

**Figure 9 sensors-25-01315-f009:**
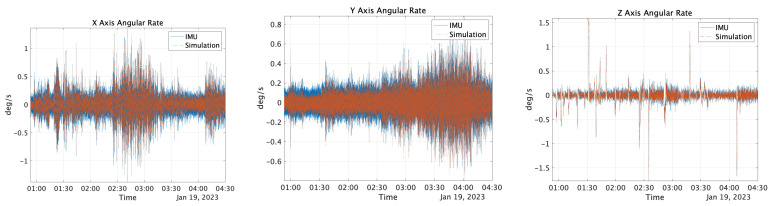
Angular rate output with IMU and simulation.

**Figure 10 sensors-25-01315-f010:**
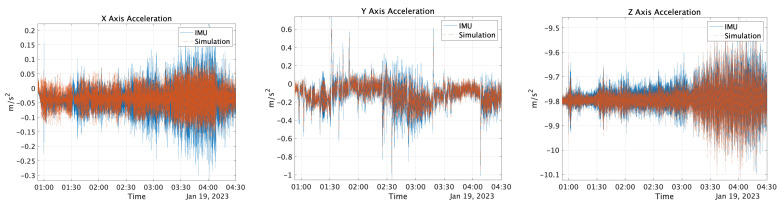
Acceleration sensor output using the IMU and simulation.

**Figure 11 sensors-25-01315-f011:**
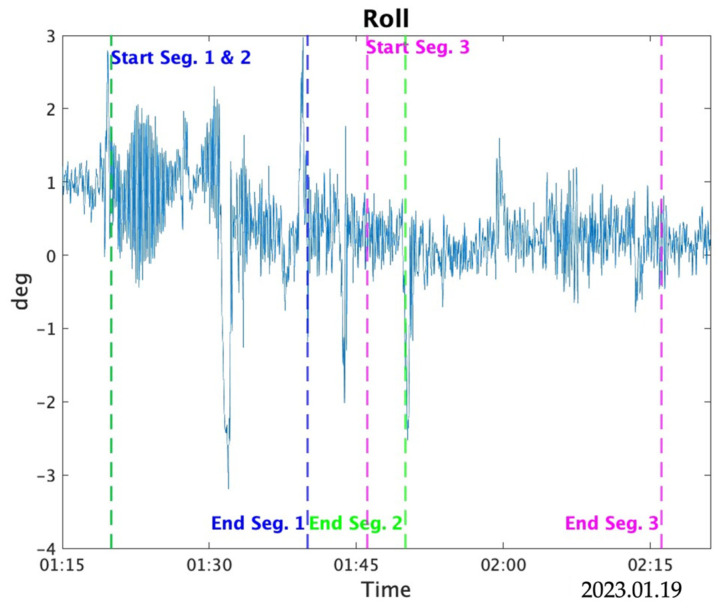
Roll with each segment.

**Figure 12 sensors-25-01315-f012:**
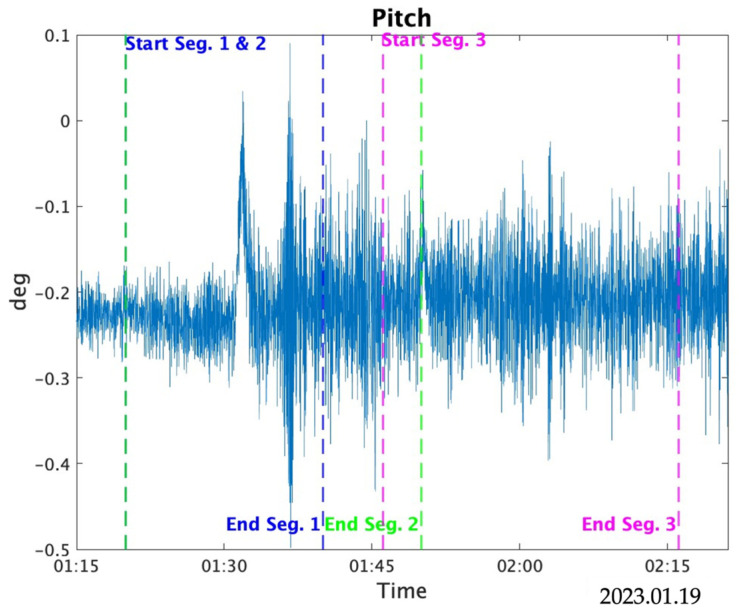
Pitch with each segment.

**Figure 13 sensors-25-01315-f013:**
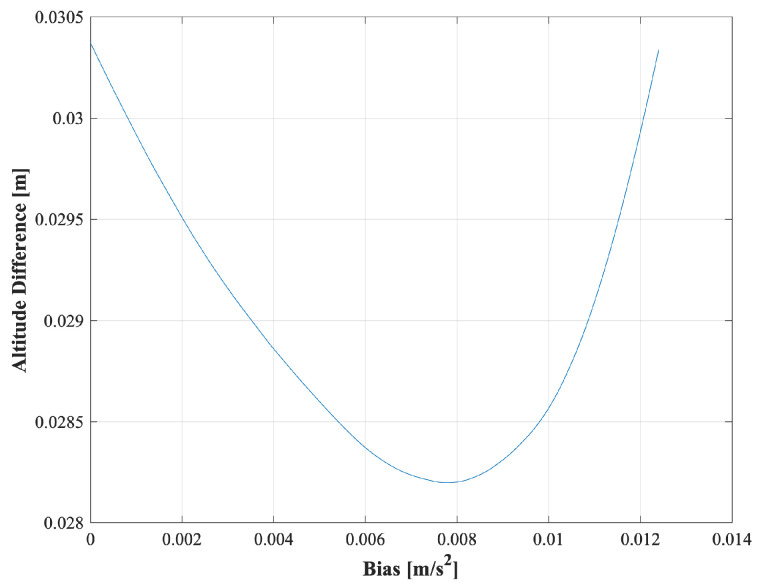
*X*-axis bias estimation.

**Figure 14 sensors-25-01315-f014:**
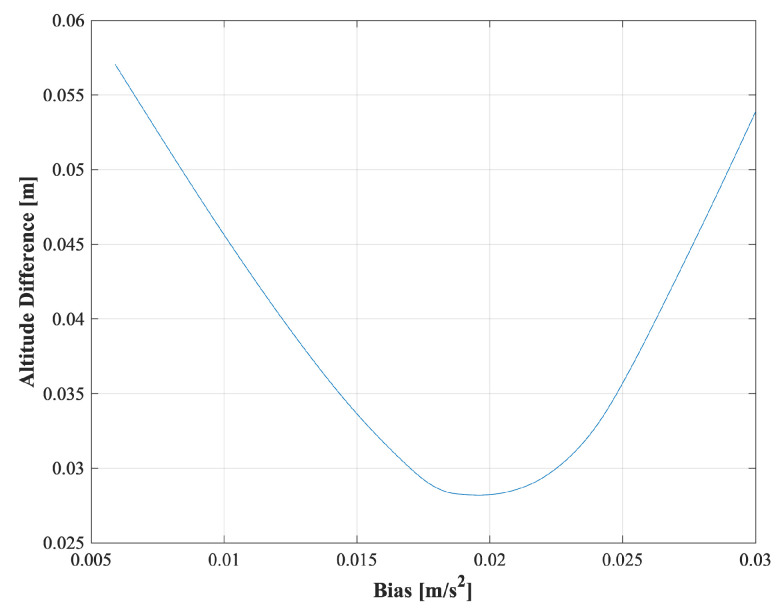
*Y*-axis bias estimation.

**Table 1 sensors-25-01315-t001:** IMU and DGNSS catalog values.

	IMU	Gyrocompass	GNSS
Company	iXblue, Saint-Germain-en-Laye, France	YDK Tec., Tokyo, Japan	S-VANS, Tokyo, Japan
Model	PHINS IMU	CMZ900	Wide-area DGNSS
Accuracy	Roll and pitch: 0.01° RMS	North-seeking	Horizontal 0.1 m
	Pure inertial drift:	Static: ±0.25 °s Lat	Vertical 0.2 m
	0.6 Nm/h	Dynamic: ±0.7 °s Lat	
TG	Roll, pitch	Heading	Position and velocity

**Table 2 sensors-25-01315-t002:** MULTI SENSOR catalog values.

	Bias	Random Walk	Bias Instability
Note	Room temperature, after warm-up		Allan variance (AV)
Accelerometer	0.0196 m/s^2^ rms	0.098 m/s^2^ p-p	-
Gyro (X and Y)	0.2 deg/s rms	0.5 deg/√h	10 deg/h
Gyro (Z)	2.78 × 10^−5^ deg/s rms	0.01 deg/√h	0.1 deg/h

**Table 3 sensors-25-01315-t003:** Parameter estimation using AV.

	STD (deg/s)(m/s^2^)	Random Walk (deg/s$\sqrt{Hz}$)(m/s^2^$\sqrt{Hz}$)	Bias Instability (deg/s)(m/s^2^)
Gyro X	4.485 × 10^−2^	6.423 × 10^−3^	5.334 × 10^−3^
Gyro Y	3.998 × 10^−2^	5.968 × 10^−3^	4.922 × 10^−3^
Gyro Z	1.371 × 10^−3^	1.882 × 10^−4^	2.191 × 10^−5^
Acc X	1.920 × 10^−3^	2.063 × 10^−4^	2.534 × 10^−5^
Acc Y	2.412 × 10^−3^	2.958 × 10^−4^	3.073 × 10^−5^
Acc Z	2.526 × 10^−3^	3.013 × 10^−4^	7.848 × 10^−5^

**Table 4 sensors-25-01315-t004:** Parameters of the GNSS simulation.

Position STD	Lat: 0.02 m	Lon: 0.02 m	Alt.: 0.05 m
Velocity STD	North: 0.0514 m/s	East: 0.0514 m/s	Down: 0.0514 m/s

**Table 5 sensors-25-01315-t005:** Angular rate bias estimation results.

	*X*-Axis Ang. deg/s		*Y*-Axis Ang. deg/s		*Z*-Axis Ang. deg/s	
	Est.	Error	Est.	Error	Est.	Error
Average	0.20	0.00	0.20	0.00	2.050 × 10^−5^	7.300 × 10^−6^
Median	0.20	0.00	0.20	0.00	1.168 × 10^−4^	8.900 × 10^−5^
Cross-Correlation	0.20	0.00	0.20	0.00	–5.809 × 10^−5^	8.589 × 10^−5^
Least Square	0.20	0.00	0.20	0.00	2.050 × 10^−5^	7.300 × 10^−6^
FFT	0.20	0.00	0.20	0.00	2.050 × 10^−5^	7.300 × 10^−6^
Butterworth	0.20	0.00	0.20	0.00	2.766 × 10^−5^	1.400 × 10^−7^

**Table 6 sensors-25-01315-t006:** Acceleration initial bias estimation results.

	*X*-Axis Acc. m/s^2^		*Y*-Axis Acc. m/s^2^		*Z*-Axis Acc. m/s^2^	
	Est.	Error	Est.	Error	Est.	Error
Average	–6.733 × 10^−3^	2.633 × 10^−2^	1.040 × 10^−2^	9.200 × 10^−3^	1.977 × 10^−2^	1.700 × 10^−4^
Median	–7.017 × 10^−3^	2.662 × 10^−2^	1.067 × 10^−2^	8.930 × 10^−3^	1.963 × 10^−2^	3.000 × 10^−5^
Cross-Correlation	–8.010 × 10^−3^	2.761 × 10^−2^	1.030 × 10^−2^	9.300 × 10^−3^	1.398 × 10^−1^	1.202 × 10^−1^
Least Square	–6.733 × 10^−3^	2.633 × 10^−2^	1.040 × 10^−2^	9.200 × 10^−3^	1.977 × 10^−2^	1.700 × 10^−4^
FFT	–6.733 × 10^−3^	2.633 × 10^−2^	1.040 × 10^−2^	9.200 × 10^−3^	1.977 × 10^−2^	1.700 × 10^−4^
Butterworth	–6.763 × 10^−3^	2.636 × 10^−2^	1.044 × 10^−2^	9.160 × 10^−3^	1.977 × 10^−2^	1.700 × 10^−4^

**Table 7 sensors-25-01315-t007:** Roll and pitch in the data interval.

	Roll (°)		Pitch (°)	
	Max	Mean	Max	Mean
20 min Seg. 1	3.19	0.62	0.50	−0.22
30 min Seg. 2	3.19	0.49	0.50	−0.22
30 min Seg. 3	2.53	0.17	0.4	−0.21

**Table 8 sensors-25-01315-t008:** 20 min data analysis results with segment 1.

Iteration	1st		2nd		3rd		4th	
*X*-Axis Bias [m/s^2^]	(0)	0.0157	(0.0157)	0.0163	(0.0163)	0.0163	(0.0163)	0.0163
*Y*-Axis Bias [m/s^2^]	0.0182	(0.0182)	0.0188	(0.0188)	0.0188	(0.0188)	0.0188	(0.0188)

**Table 9 sensors-25-01315-t009:** 30 min data analysis results with segment 2.

Iteration	1st		2nd		3rd		4th	
*X*-Axis Bias [m/s^2^]	0	0.0072	0.0072	0.0078	0.0079	0.0079	0.0079	0.0079
*Y*-Axis Bias [m/s^2^]	0.0207	0.0207	0.0197	0.0197	0.0197	0.0196	0.0196	0.0195

**Table 10 sensors-25-01315-t010:** 30 min data analysis results with segment 3.

Iteration	1st		2nd		3rd		4th	
*X*-Axis Bias [m/s^2^]	0.027	(0.027)	0.022	(0.022)	0.0219	(0.0219)	0.0219	(0.0219)
*Y*-Axis Bias [m/s^2^]	(0)	0.0123	(0.0123)	0.0126	(0.0126)	0.0127	(0.0127)	0.0127

## Data Availability

The data presented in this study are available upon request from the corresponding author. The data were not publicly available because of the training ship. When a request is received for a reasonable reason, it can only be provided after explaining it to the relevant department and obtaining permission from all departments.

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
