# Peer review of "Bias Estimation for Low-Cost IMU Including X- and Y-Axis Accelerometers in INS/GPS/Gyrocompass"

_sensors, 2025, doi:10.3390/s25051315_

Round 1

Reviewer 1 Report

Comments and Suggestions for Authors

Low-cost MEMS IMUs suffer from significant bias errors, which degrade inertial navigation accuracy, especially in GNSS-denied maritime environments. Existing bias estimation methods struggle with maritime applications due to continuous vessel motion and the inability to apply standard techniques like Zero Velocity Update (ZUPT). A trajectory generator (TG)-based method is introduced to estimate IMU bias by comparing estimated and measured acceleration/angular velocity values. And novel altitude-based bias estimation method is proposed for X- and Y-axis accelerometers, leveraging altitude differences between INS/GNSS/gyrocompass (IGG) and TG-estimated values. While promising, real-time implementation and theoretical justification for the altitude-based method remain as future challenges.

Before the paper is published, some details can be revised:

  1. In Table 3, Figures 7, and 8, what was the sampling duration of the IMU data used for Allan variance analysis? What conclusions were drawn from the Allan variance analysis? How did these findings contribute to subsequent experiments? This section requires further discussion.
  2. The paper previously stated that the proposed method enhances the performance of low-cost MEMS IMUs in GNSS-denied environments. However, the study only briefly touches on this aspect theoretically and primarily validates IMU error estimation through simulation experiments. It does not assess actual performance under GNSS-denied conditions. Therefore, the initial claim is inaccurate and should be revised accordingly. Especially in the abstract.
  3. In practical applications, MEMS IMU bias can be highly unstable and influenced by multiple factors. Can the applicability of the proposed method and conclusions be discussed in relation to different grades of IMUs?

Furthermore,this paper could benefit from deeper theoretical justification, real-world validation, and benchmarking against alternative methods. If these areas are addressed, the research could have a significant impact on low-cost autonomous navigation systems.

Author Response

Thank you for your valuable comments. Below, we provide our responses to the reviewer's feedback.

Additionally, all figures in the manuscript have been revised and formatted appropriately by a professional English editing service. Furthermore, the Word document was edited with the "Track Changes" feature enabled, and all modifications and additions have been highlighted in green for clarity.

Comment 1 : In Table 3, Figures 7, and 8, what was the sampling duration of the IMU data used for Allan variance analysis? What conclusions were drawn from the Allan variance analysis? How did these findings contribute to subsequent experiments? This section requires further discussion.

Response 1: 

Thank you for your insightful comment. The Allan variance analysis was conducted for sensor modeling purposes, using IMU data sampled at 50 Hz over a duration of 44 hours. This information has been added to the manuscript. Additionally, the results obtained from the Allan variance analysis were utilized to generate simulation data, as described in the manuscript. Furthermore, the inclusion of Allan variance results serves an additional purpose: allowing other researchers to assess the accuracy of the IMU used in this study. In summary, the Allan variance analysis was performed to obtain sensor simulation values, and its results are also included to help other researchers evaluate the precision of the IMU employed in this research.

Comment 2: The paper previously stated that the proposed method enhances the performance of low-cost MEMS IMUs in GNSS-denied environments. However, the study only briefly touches on this aspect theoretically and primarily validates IMU error estimation through simulation experiments. It does not assess actual performance under GNSS-denied conditions. Therefore, the initial claim is inaccurate and should be revised accordingly. Especially in the abstract.

Response 2:

Thank you for your valuable feedback. Previous research [6] has demonstrated that the accuracy of initial bias estimation significantly affects positioning accuracy. Since this paper is an updated version of that study, we acknowledge that the evaluation of performance in GNSS-denied environments has not been conducted within this paper. However, the theoretical claims made are not incorrect. That said, we understand the concern that mentioning this aspect in the abstract may lead to potential misunderstandings for readers. Therefore, we have revised the abstract accordingly to prevent any misinterpretation.

Comment 3: In practical applications, MEMS IMU bias can be highly unstable and influenced by multiple factors. Can the applicability of the proposed method and conclusions be discussed in relation to different grades of IMUs?

Response 3:

Thank you for your insightful comment. As you pointed out, MEMS IMU bias is highly unstable and poses significant challenges for researchers. In particular, temperature-induced instability varies greatly between individual units, making it difficult to create a universal model. Instead, we believe that the best approach is to assess each unit’s response to temperature variations. Additionally, since real-world environments are constantly changing, ensuring repeatability is a significant challenge.

This is precisely why our proposed method leverages a trajectory generator (TG). Given the considerable variability in MEMS IMU bias due to environmental factors, modeling such biases is inherently difficult. Instead, our approach determines the acceleration and angular velocity that minimize position estimation errors when using pure inertial navigation and aligns the measured values accordingly through a TG-based system. From this perspective, we believe that our method is applicable to different grades of IMUs. In fact, we have confirmed accuracy improvements using three different IMUs in our experiments.

However, given the characteristics of MEMS sensors discussed above, we acknowledge that it is difficult to generalize or definitively claim that this method can be applied to all grades of MEMS IMUs. As a conclusion, rather than expanding the discussion on different IMU grades within this paper, we believe it is more important for individual researchers to apply this method to their own IMUs and evaluate its effectiveness.

Additional Comment : Furthermore,this paper could benefit from deeper theoretical justification, real-world validation, and benchmarking against alternative methods. If these areas are addressed, the research could have a significant impact on low-cost autonomous navigation systems.

Response:

Thank you very much for your encouraging comments. We truly appreciate your valuable feedback. We will certainly conduct validation using real-world measurements. The key challenge will be whether the theoretical aspects can be fully substantiated, but we are motivated by the desire to contribute to this field in any way we can.

Reviewer 2 Report

Comments and Suggestions for Authors

REVIEW OF

Bias Estimation for Low-Cost IMU Including X- and Y-Axis Accelerometers in INS/GPS/Gyrocompass

BY

Gen Fukuda and Nobuaki Kubo

The article is devoted to an urgent modern task. In a broad sense, this is positioning without global navigation. There are a lot of such studies, but the authors solve a much narrower problem. In fact, they limit the cost of navigation equipment and get the simplest and cheapest system, which they explore. The INS/GNSS/gyrocompass system is shown in Figure 2. The task is to determine the bias typical for low-cost IMU devices. The method consists in modeling the trajectories and calculating the correlation observed by the authors using the example of real experimental data.

The article is well written, well-structured and designed. A great experimental section. The article can be published. There are a few small comments.

  1. Authors should specify ORCIDs and ensure that the completed profiles are complete, so that the reader can easily get an idea of the authors' previous work and competencies.
  2. The abbreviation in the list of keywords should be represented by the full name.
  3. The performed research is based on an experimental (engineering) analysis of the patterns of a very specific observation system. There are no guarantees that the situation will repeat in other models and the methods proposed by the authors will work. What should I do about it?
  4. The reviewer found no clear explanation for the difference in ratings on the X and Y axes. If the Z-axis differences are obvious, then there doesn't seem to be a physical difference between X and Y. Wouldn't turning the accelerometer 90 degrees change the positions of X and Y? So why the difference?
  5. There is an assumption that a completely different approach to the problem as a whole is possible. Precisely, if we abandon the use of the simplest filter (Extended Kalman Filter) for data integration and use more subtle approaches, then the problem of bias estimation will go away by itself, since it will be solved by an effective filtering algorithm, rather than involving engineering observations based on technical analysis of rather modest data sets. The point is to take into account the bias directly in the observation system model and "entrust" its evaluation to the filter. EKF cannot and will not cope with such a task. Hence the research, including the authors in this and previous works. More advanced filters can handle this task without engineering tricks. Thus, I would like to see the answer to the question in the text: is there a place for more advanced mathematics in this work?

Author Response

We sincerely appreciate the reviewer’s valuable comments and feedback. Below, we provide detailed responses to each of the points raised. Additionally, all figures have been revised and formatted according to the journal’s requirements by a professional English editing service. Furthermore, the Track Changes feature in Microsoft Word has been enabled during the revision process. All modifications and additions are highlighted in green for clarity.

Comment 1: Authors should specify ORCIDs and ensure that the completed profiles are complete, so that the reader can easily get an idea of the authors' previous work and competencies.

Response 1:

Thank you for your suggestion regarding ORCIDs. Our previously published papers are openly available, allowing readers to review our past research and areas of expertise. Each author has reviewed their profile and will add relevant information where possible. However, as we do not actively maintain our ORCIDs with extensive details, we kindly ask for flexibility regarding the extent of the updates. Additionally, both authors actively use ResearchGate, where more information about our research can be found.

Comment 2: The abbreviation in the list of keywords should be represented by the full name.

Response 2:

We have replaced "MEMS" with "micro-electro-mechanical system" throughout the manuscript.

Comment 3: The performed research is based on an experimental (engineering) analysis of the patterns of a very specific observation system. There are no guarantees that the situation will repeat in other models and the methods proposed by the authors will work. What should I do about it?

Response 3:

Thank you for your insightful comment. Based on previous research, we have demonstrated the effectiveness of trajectory generator (TG)-based methods for INS applications. From this perspective, if researchers develop their own TG for INS calculations and apply similar methodologies, we believe that the proposed method can be reproduced.

Furthermore, our previous studies, published in The Journal of Navigation and Sensors[6,29], have already validated the estimation of acceleration and angular velocity using TG-derived measurements. In the case of maritime applications, the proposed method can be replicated by other researchers using a low-cost IMU, an INS/GNSS/gyrocompass (IGG), and a TG-based acceleration and angular velocity estimation system.

However, whether the same results can be achieved using a completely different approach for acceleration and angular velocity estimation depends on the outcomes of that specific research. Therefore, we are unable to make a definitive judgment on this matter.

Comment 4: The reviewer found no clear explanation for the difference in ratings on the X and Y axes. If the Z-axis differences are obvious, then there doesn't seem to be a physical difference between X and Y. Wouldn't turning the accelerometer 90 degrees change the positions of X and Y? So why the difference?

Response 4:

Thank you for your insightful question. As you pointed out, there is no physical difference between the X and Y axes when the system is in a perfectly horizontal state. However, when roll and pitch motions are introduced, the behavior of each axis changes.

Specifically, rolling around the X-axis affects the Y-axis acceleration, while pitching around the Y-axis affects the X-axis acceleration. As mentioned in multiple sections of the manuscript, these roll- and pitch-induced acceleration inputs are essential for estimating the initial bias of the X- and Y-axis accelerations.

The simplest bias estimation method involves precisely rolling the system while stationary to measure the gravity acceleration component and determine the initial bias of the Y-axis acceleration. However, achieving perfect rolling is extremely challenging. In maritime applications, additional ship-induced rolling must also be considered, making it impractical to mount the IMU on equipment capable of performing such controlled motions. The same challenges apply to X-axis acceleration estimation.

While the X and Y axes are physically identical, the influence of roll and pitch leads to differences in initial bias estimation accuracy between the two axes.

Comment 5: There is an assumption that a completely different approach to the problem as a whole is possible. Precisely, if we abandon the use of the simplest filter (Extended Kalman Filter) for data integration and use more subtle approaches, then the problem of bias estimation will go away by itself, since it will be solved by an effective filtering algorithm, rather than involving engineering observations based on technical analysis of rather modest data sets. The point is to take into account the bias directly in the observation system model and "entrust" its evaluation to the filter. EKF cannot and will not cope with such a task. Hence the research, including the authors in this and previous works. More advanced filters can handle this task without engineering tricks. Thus, I would like to see the answer to the question in the text: is there a place for more advanced mathematics in this work?

Response 5:

Thank you for your insightful comment. The difficulty in estimating the initial bias of the X- and Y-axis accelerometers arises from the fact that acceleration sensors cannot distinguish between motion-induced acceleration and gravity-induced acceleration. To address this, previous studies[14,15] have utilized the fact that the system is stationary to intentionally apply gravity acceleration components through rolling and pitching motions, allowing for initial bias estimation using simultaneous equations.

Furthermore, in the case of low-cost MEMS-based IMUs, the initial bias is affected by factors such as temperature variations, making it difficult to model accurately. To the best of our knowledge, no prior research has successfully established a robust modeling approach for this phenomenon. However, if MEMS-based IMUs achieve higher precision in the future, alignment methods currently used in inertial navigation systems employing FOG or RLG may provide a viable solution.

If an accurate model of the initial bias behavior of MEMS-based IMUs can be established, mathematical methods may indeed offer a solution for bias estimation. Additionally, if we can mathematically demonstrate the correlation between the altitude difference observed between IGG and TG and the bias values identified in our approach, then a purely mathematical solution could be feasible. However, as noted in the manuscript, we have not yet been able to mathematically validate this correlation. Therefore, at this stage, it remains uncertain whether advanced mathematical techniques are necessary for this problem.
